# Sleep Debt Mediates the Relationship between Work-Related Social Factors, Presenteeism, and Well-Being in Japanese Workers

**DOI:** 10.3390/ijerph20075310

**Published:** 2023-03-29

**Authors:** Yuta Takano, Suguru Iwano, Takeshi Ando, Isa Okajima

**Affiliations:** 1Department of Psychology, Faculty of Human Culture and Sciences, Fukuyama University, Hiroshima 729-0292, Japan; 2Graduate School of Psychological Science, Health Sciences University of Hokkaido, Hokkaido 002-8072, Japan; 3Faculty of Welfare and Health Science, Oita University, Oita 870-1192, Japan; 4Department of Psychological Counseling, Faculty of Humanities, Tokyo Kasei University, Tokyo 173-8602, Japan

**Keywords:** commute, employee, psychological well-being, sleep, work productivity

## Abstract

Sleep debt is associated with presenteeism and mental health; however, the association of sleep debt with presenteeism and well-being in the context of work-related social factors (commuting time, workdays, and working hours) has not been fully elucidated. This study aimed to examine whether work-related social factors are associated with presenteeism and well-being via sleep debt. The participant group comprised 872 full-time and 526 part-time workers (mean age: 44.65 ± 12.37 and 48.47 ± 12.93 years, respectively). For both the full-time and part-time workers, increased sleep debt was significantly associated with presenteeism (β = −0.171; β = −0.160) and low well-being (β = −0.135; β = −0.153). Notably, commuting time was significantly associated with increased sleep debt in full-time workers (β = 0.09). In contrast, the number of workdays was significantly associated with increased sleep debt in part-time workers (β = −0.102). Working hours were not significantly associated with sleep debt for both full- and part-time workers. These results reveal that sleep debt might lead to various risks among workers, elucidating the work-related social factors related to sleep debt. They also highlight the importance of considering work-related social factors when addressing sleep debt.

## 1. Introduction

Sleep duration is considered to have a U-shaped curve in relation to presenteeism and mental health. Presenteeism is defined as the state of being at work but experiencing a loss of productivity due to health problems [1], and short and long sleep durations are more likely to result in presenteeism than a normal sleep duration [2,3]. Short and long sleep durations have also been shown to be associated with depression [4]. However, classification by absolute sleep duration may not reflect loss from the amount of sleep required by an individual; therefore, it is also necessary to evaluate sleep debt [5], which is linearly related to presenteeism and mental health [5,6]. As sleep duration was not appropriate for addressing correlations because of the U-shaped curve in relation to presenteeism and mental health, and because this U-shaped curve problem could be solved by using sleep debt [5], in the present study, sleep debt was used as a short sleep indicator.

Short sleep durations are associated with several factors; the most representative are working hours and commuting time [7]. For example, long working hours per day were found to be associated with short sleep durations (<6 h) in both men and women [8], and individuals who worked more than 40 h per week had a shorter sleep duration compared with those who did not [9]. Considering both daily and weekly working hours as relevant, it is necessary to consider the number of workdays per week. Several studies have also investigated the relationship between sleep duration and commuting time, which is associated with objective sleep duration as measured using actigraphy [10]. Sleep duration is the most influential factor in the association between commuting time and health-related activities such as physical activity, meal preparation, mealtime, and optimal sleep duration [11]. These findings suggest that working hours, workdays, and commuting times reduce sleep duration, which might be associated with sleep debt. Thus, work-related social factors such as working hours, commuting times, and workdays may be associated with increased sleep debt.

Sleep debt is associated with not only presenteeism but also mental health; therefore, it might also be associated with low well-being. On the other hand, since presenteeism is a state of loss of productivity due to health problems [1], a relationship between presenteeism and well-being mediated by sleep debt, rather than direct relationships between work-related factors, is expected. Previous studies have examined the relationships between individual variables but have not comprehensively examined the social phenomenon of work-related social factors, presenteeism, and well-being from the perspective of sleep debt [5,6,7,8,9,10,11]. The COVID-19 pandemic caused lockdowns worldwide. Initially, it was believed that the lockdown would cause or exacerbate sleep problems, but several studies have reported that they rather increased sleep duration [12,13]. During the lockdown, most people worked from home, and that condition was associated with worse presenteeism [14]. That is, the results indicate that even with signs of improvement in sleep debt due to lifestyle changes, improvement in job performance may not be immediately apparent. Furthermore, no model comprehensively examines work-related social factors, sleep debt, presenteeism, and well-being caused by the lifestyle changes after the outbreak. It is important to understand the relationship between work-related social factors, sleep debt, presenteeism, and well-being to optimize worker interventions.

The 2021 Labor Force Survey found that 36.7% of the Japanese working population are part-time workers, with the most common reason for this being a desire to work at their convenience [15]. Hence, work-related social factors associated with sleep debt may differ between a part-time and full-time status.

This study aimed to examine whether working hours, workdays, and commuting times are associated with presenteeism and well-being via sleep debt in full-time and part-time workers using structural equation modeling (SEM). The hypothetical model for this study is presented in Figure 1.

## 2. Materials and Methods

### 2.1. Procedures and Participants

This study was approved by the Research Ethics Committee of Fukuyama University, Japan (approval number: 2022-H-15). All procedures were performed in accordance with the ethical standards of the Institutional and National Research Committee and the 1964 Helsinki Declaration. Informed consent was obtained from all participants.

The screening criteria were an age of <70 years and being a worker. The Directed Questions Scale [16] was used to check for inattention; two inattention detection items were used in this study. The exclusion criteria were inaccurate answers to inattention detection items, working from home at least 3 days per week, going to work less than 3 days per week, having no commuting time, and having a negative Sleep Debt Index (SDI) value [5]. In total, 1398 participants (660 males, 731 females, and 7 people of other genders) of a mean age of 46.09 ± 12.72 years were included. The participant selection process is shown in Figure 2.

The study survey was conducted by an Internet research company (Cross Marketing, Inc., Tokyo, Japan). Cross Marketing’s active panel for the most recent year was 2.95 million. The survey was randomly administered to those aged 20–69 years registered as workers of an Internet survey company (Cross Marketing, Inc.). Those who agreed to participate in the survey were directed to the Qualtrics form. Qualtrics is an Internet survey tool, and its optional features were set as follows: no back button, no resumption after interruption in the middle of an answer, no multiple answers for the same respondent, and bot detection. The purpose of the survey and ethical considerations were presented in writing on the screen before the questionnaire items. The survey was conducted between 1 June 2022, and 8 June 2022.

### 2.2. Measures

The demographic data were age, gender, marital status, employment status, occupation, number of workdays, and weekly working hours. Occupational classification was performed according to the Japan Standard Occupational Classification [17]. Working hours were self-reported and assessed based on the question, “How long do you work in a week?” Commuting time was also self-reported and assessed based on the question, “How long is your daily round-trip commuting time?” Weekly commuting time was calculated by multiplying the daily commuting time by the number of workdays.

The SDI is a validated self-report scale for assessing sleep debt [5]. The SDI consists of three questions: (1) How long did you sleep at night during workdays in the last month? (2) How long did you sleep on the day-off in the last month? (3) Considering your own “feeling best performance” rhythms, for how long would you sleep if you were entirely free for the day? The weekday and day-off total sleep times in the last month were recorded, and actual total sleep time was calculated by adding the weekday and weekend total sleep times and dividing the answer by 7. The SDI value was calculated by subtracting the actual total sleep time from the ideal total sleep time. Weekdays were defined as the number of workdays, and days off were calculated by subtracting the number of workdays from 7. Increased sleep debt, as assessed using the SDI, is associated with increased depressive symptoms, sleepiness, and reduced work performance. Moreover, a higher SDI value indicates a greater sleep debt [5].

The short-form Japanese version of the World Health Organization Health and Work Performance Questionnaire (WHO-HPQ) is a self-report questionnaire that assesses presenteeism and absenteeism [18]. It assesses presenteeism as absolute and relative. Kessler et al. [19] determined that the self-evaluation of absolute presenteeism was consistent with the supervisor’s evaluation of the individual’s work performance in the calibration study for the WHO-HPQ. Thus, the current study used only one absolute presenteeism item: “How would you rate your overall job performance on the days you worked during the past 4 weeks (28 days)?” Presenteeism was scored on an 11-point Likert scale ranging from 0 (worst performance) to 10 (top performance). The absolute presenteeism score was calculated by multiplying the raw score by 10. The short-form Japanese version of the WHO-HPQ is associated with depressive symptoms at 1 year and has predictive validity [20]. A high WHO-HPQ score indicates good work performance, and a low score indicates presenteeism.

The brief version of the Psychological Well-Being Scale (PWBS) is a validated self-report scale for assessing psychological well-being [18]. The PWBS consists of 24 items based on the item response theory and reflects the concept of psychological well-being. PWBS is scored on a 6-point Likert scale ranging from 1 (never) to 6 (always). This score has a high-order factor structure and good test–retest reliability (*r* = 0.85) after 1 month [21] and is positively correlated with happiness and positive affect and negatively correlated with depressive symptoms and negative affect [21]. Hence, a higher PWBS score indicates a higher level of well-being.

### 2.3. Statistical Analyses

Welch’s *t*-test was used to examine whether there were significant differences in age, working hours per week, commuting time per week, sleep debt (SDI values), presenteeism (WHO-HPQ scores), and psychological well-being (PWBS scores) between full-time and part-time workers. Hedges’ *g* was used to calculate effect sizes; an effect size from ≥0.2 to <0.5 was classified as a small effect size, an effect size from ≥0.5 to <0.8 was classified as a moderate effect size, and ≥0.8 was classified as a large effect size. Hedges’ *g* was calculated using the R package “compute.es” (version: 0.2.5). A 95% confidence interval (CI) was calculated for both the *t*-test and Hedges’ *g*.

Pearson’s product–moment correlation analyses were conducted to examine the association between workdays per week, working hours per week, commuting time per week, sleep debt (SDI values), presenteeism (WHO-HPQ scores), and psychological well-being (PWBS scores) for the full- and part-time workers. Correlation coefficient values were calculated using the R package “corrplot” (version: 0.92). A 95% CI was calculated for the correlation coefficient values.

SEM was conducted for both full-time and part-time workers according to our hypothesized model (Figure 1) using the R package “lavaan” (version: 0.6.7), and the maximum likelihood method was used for estimation. This study evaluated the chi-square result ($\chi^{2}$), comparative fit index (CFI), root mean square error of approximation (RMSEA), goodness of fit index (GFI), and adjusted GFI (AGFI) as indicators of model fit. Model fit indicators were evaluated as good for a CFI of ≥0.95, a RMSEA of <0.08, and GFI and AGFI values of ≥0.95 [22]. The R package “semTools” (version: 0.5.6) was used to calculate the sample size with 6 degrees of freedom; α = 0.05, and 1-β = 0.8. The null hypothesis of RMSEA was set to zero, and the alternative hypothesis was set to 0.08, resulting in a sample size of 356 participants. Therefore, we recruited 356 participants or more in each group.

All statistical analyses were performed using R software version 3.6.1 and version 4.2.2 (The R Project for Statistical Computing).

## 3. Results

The demographic data of the participants are presented in Table 1.

The results of Welch’s *t*-tests for group differences between the full-time and part-time workers indicated that the full-time workers were significantly younger (*t* (1068.5) = 5.44 (95% CI: 2.44, 5.20)) and had significantly longer weekly workdays (*t* (741.84) = −14.87 (95% CI: −0.67, −0.51)), weekly working hours (*t* (1264) = −15.52 (95% CI: −14,26, −11.06)), and weekly commuting times (*t* (1305.4) = −9.60 (95% CI: −2.42, −1.60)). They also had significantly lower WHO-HPQ scores (*t* (1054.5) = 2.18 (95% CI: 0.23, 4.36)) and significantly higher PWBS scores (*t* (1048.7) = −2.32 (95% CI: −3.95, −0.33)). There were no significant differences between both groups in SDI values (*t* (1064.2) = −1.57 (95% CI: −0.21, 0.02)). The mean, standard deviation, and effect size values for age, weekly working hours, weekly commuting time, and scores on each scale are presented in Table 2.

The correlation coefficient values for the associations between the variables in the SEM are shown in Table 3. The SEM was conducted according to the hypothesized model shown in Figure 1. The path coefficients are listed in Table 4. First, the model fit and path coefficients were estimated for each variable. The results indicated that the model was a good fit for all variables (χ^2^ (6) = 15.372, *p* = 0.018; CFI = 0.973, RMSEA = 0.033, GFI = 0.993, and AGFI = 0.975). The path coefficients showed that increases in workdays and commuting times were significantly associated with increases in SDI values (β = 0.072, *p* = 0.012; β = 0.077, *p* = 0.004). Meanwhile, working hours were not significantly associated with SDI values (β = −0.013, *p* = 0.659). Increased SDI values were significantly associated with decreased WHO-HPQ scores (β = −0.169, *p* < 0.001) and PWBS (β = −0.140, *p* < 0.001). The R^2^ values for SDI, WHO-HPQ score, and PWBS score were 0.012, 0.029, and 0.019, respectively.

Second, the model fit and path coefficients for each variable were estimated for each employment status variable. The results indicated that the model was a good fit for full-time workers (χ^2^ (6) = 3.233, *p* = 0.779; CFI = 1.00, RMSEA = 0.000, GFI = 0.998, and AGFI = 0.991). The path coefficients showed that an increased commuting time was significantly associated with an increase in the SDI value (β = 0.090, *p* = 0.008). Working hours and workdays were not significantly associated with SDI values (β = −0.006, *p* = 0.856; β = 0.038, *p* = 0.273). An increase in the SDI value was significantly associated with a decrease in the scores of the WHO-HPQ (β = −0.171, *p* < 0.001) and PWBS (β = −0.135, *p* < 0.001). The R^2^ values for the SDI, WHO-HPQ score, and PWBS score were 0.009, 0.029, and 0.018, respectively.

The results also indicated that the model was a good fit for part-time workers (χ^2^ (6) = 16.934, *p* = 0.010; CFI = 0.930, RMSEA = 0.059, GFI = 0.979, and AGFI = 0.927). Path coefficients showed that an increase in workdays was significantly associated with the SDI values (β = 0.102, *p* = 0.028). Working hours and commuting time were not significantly associated with SDI value (β = −0.020, *p* = 0.664; β = 0.045, *p* = 0.317). An increase in SDI value was significantly associated with decreased WHO-HPQ (β = −0.160, *p* < 0.001) and PWBS (β = −0.153, *p* < 0.001) scores. The R^2^ values for the SDI, WHO-HPQ score, and PWBS score were 0.013, 0.026, and 0.024, respectively.

## 4. Discussion

The association of sleep debt with presenteeism and well-being in the context of work-related social factors has not been fully elucidated. This study found that increased sleep debt was associated with presenteeism and low well-being in full-time and part-time workers. However, the associations between sleep debt and working hours, workdays, and commuting time differed between the full-time and part-time workers. Only increased commuting time was associated with increased sleep debt for the full-time workers. In contrast, only increased workdays were associated with increased sleep debt for the part-time workers. Working hours were not associated with sleep debt for both full- and part-time workers.

The results, indicating that sleep debt is associated with presenteeism and low well-being, are consistent with those of previous studies [5,6]. The novelty of the current study results lies in the demonstration of various risks posed by sleep debt regardless of a full-time or part-time employment status. A systematic review of the effects of sleep restriction on cognitive performance showed that worse performance in vigilance and simple attentional tasks was a robust outcome [23]. When the sleep duration was 8 h in the control condition and 5.6 h in the sleep restriction condition, cognitive performance was 15.9% lower in the sleep restriction condition than in the control condition immediately after waking [24]. Furthermore, at 70 min after waking, cognitive performance was still lower in the sleep restriction condition than in the control condition. Moreover, although subjective daytime sleepiness was comparable under both conditions [24], subjective sleepiness was perceived less under sleep restriction conditions, despite the reduced cognitive performance [25]. These observations indicate that although increased sleep debt might impair cognitive performance, making it impossible to work efficiently and complete the expected workload, it is difficult to determine whether individuals are obtaining sufficient sleep because they cannot recognize sleepiness during the day. Therefore, the effect of sleep debt on presenteeism may not be recognized immediately.

Sleep debt is associated with presenteeism and low well-being. A meta-analysis found that an objective short sleep duration (sleep deprivation or sleep restriction) increased negative mood and decreased positive mood [26]. Given that sleep deprivation increased negative mood more than sleep restriction but did not make a difference in positive mood, Tomaso et al. [26] argued that even a few hours of sleep restriction can affect positive mood as much as not sleeping. The results of a crossover study in which participants were assigned to a constant sleep condition for 5 days showed that participants with the normal sleep conditions of 7–8 h of sleep and those with sleep restrictions under which the maximum sleep duration was 5 h did not differ in their evaluation of unpleasant pictures. However, participants with sleep restriction negatively evaluated pleasant and neutral pictures; the results did not change when mood states were controlled for [27]. Hence, increased sleep debt is associated with low well-being. Additionally, increased sleep debt may negatively affect interpersonal communication, decreasing work efficiency.

In the current study, work-related social factors related to sleep debt differed according to whether the participants were of a full-time or part-time employment status. Full-time workers were associated with longer commuting times and higher sleep debt, which is consistent with previous findings [7,11]. Meanwhile, part-time workers were associated with more workdays and higher sleep debt but not longer commuting times. This could have been due to a difference of approximately 2 h per week in commuting times between full-time workers and part-time workers. Workers choose to work part-time because they want to work at their convenience [15]. Hence, there may be a difference in the factors related to sleep debt between full-time and part-time workers. Although working hours have been associated with short sleep durations [7], working hours were not associated with sleep debt in the current study. Additionally, a cohort study in the United Kingdom found that although working hours were associated with short sleep durations, there was no significant difference between working for 41–55 h and working for 35–40 h [28]. The average weekly working hours in this study were 34.44 ± 16.47 h for full-time workers and 21.78 ± 13.66 h for part-time workers. The average weekly working hours being less than 40 could explain the lack of association between working hours and sleep debt. Further studies of the association between working hours and sleep debt are required. A systematic review that examined whether interventions for sleep problems improved presenteeism found no support programs addressing sleep debt [29]. Furthermore, having a shorter amount of time between the end of a workday and the next workday increased sleep debt, but an extended time did not decrease sleep debt [30]. Therefore, support programs are needed to address sleep debt better. For example, employee dormitories may need to be built closer to the workplace, or employees could live closer to work; the number of workdays should also not be increased.

This study has several limitations. First, this was a cross-sectional study, and causal relationships could not be determined. Therefore, the results of this study were limited to the associations between work-related social factors and sleep debt and the associations among sleep debt, presenteeism, and well-being. Second, it should be cautioned that this study was conducted on individuals registered with an Internet research company, although 37.6% of the respondents were part-time workers, which is consistent with a representative sample in Japan [15]. Third, reporting bias was possible because this study was based on self-report scales. In future studies, an objective evaluation using actigraphy will be necessary to assess sleep debt better. Fourth, this study was conducted on Japanese workers. The relationship between commuting time and sleep debt has been reported in the US [7,10,11] and Sweden [31]. On the other hand, the U.S. Bureau of Labor Statistics reported that among Americans, “Other family and/or personal obligations” are the most common reasons for part-time work choices among 25 to 64 years old [32]. In the future, it is necessary to examine whether our model can be replicated in other countries. Fifth, other work-related social factors such as job position, income, and the mode of commuting were not included in this study. In particular, as a commuting mode, the combination of riding a bicycle and walking is associated with a higher level of well-being compared to driving a car [33].

## 5. Conclusions

In conclusion, this study captured the increasingly significant social phenomena of commuting time, number of workdays, sleep debt, presenteeism, and well-being. For the first time, our study found that factors associated with sleep debt differ based on whether someone is of a full-time or part-time employment status. The findings showed that reducing working hours does not generally eliminate sleep debt. This means that people of different employment statuses require different methods to address sleep debt. In situations such as the COVID-19 pandemic, during which lifestyles change remarkably, the impact of the change on workers’ sleep may likely vary by job type and employment status.

## Figures and Tables

**Figure 1 ijerph-20-05310-f001:**
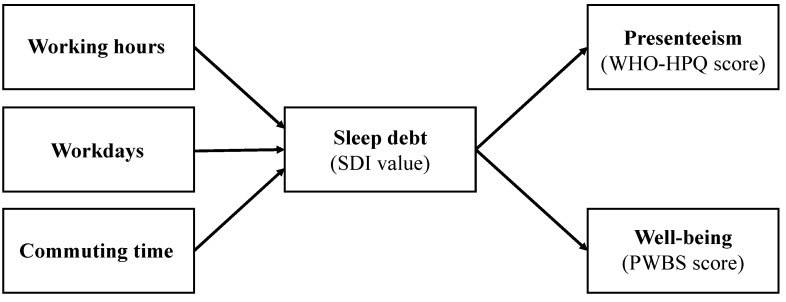
Hypothetical model for this study. PWBS, Psychological Well-Being Scale; SDI, Sleep Debt Index; WHO-HPQ, World Health Organization Health and Work Performance Questionnaire.

**Figure 2 ijerph-20-05310-f002:**
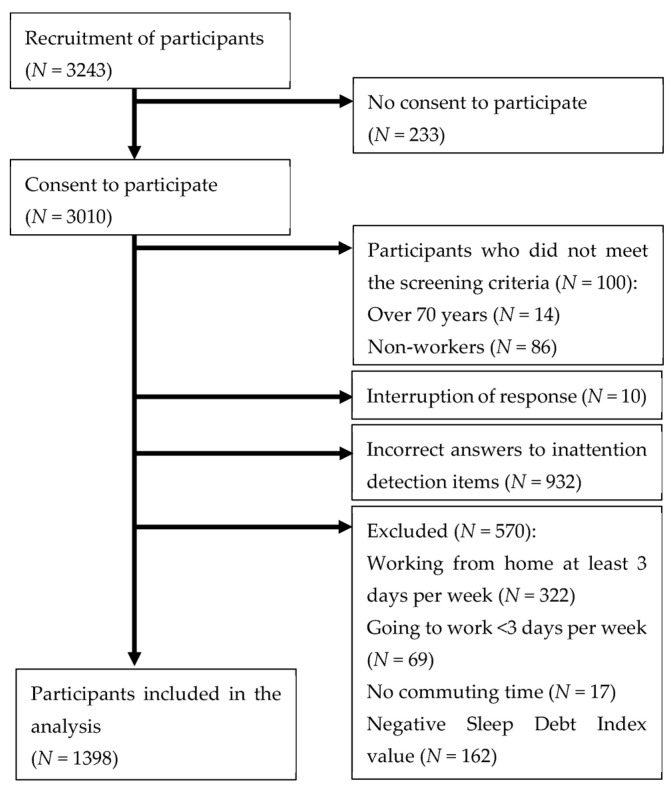
Flowchart of participant selection.

**Table 1 ijerph-20-05310-t001:** Sociodemographic data of the participants.

	All Workers (*N* = 1398)	Full-Time Workers (*N* = 872)	Part-Time Workers (*N* = 526)
Gender, *n* (%)			
Male	660 (47.2)	347 (39.8)	384 (73.0)
Female	731 (52.3)	520 (59.6)	140 (26.6)
Other	7 (0.5)	5 (0.6)	2 (0.4)
Marital status, *n* (%)			
Single	641 (45.9)	407 (46.7)	234 (44.5)
Married	757 (54.1)	465 (53.3)	292 (55.5)
Occupation, *n* (%)			
Administrative and managerial	101 (7.2)	98 (11.2)	3 (0.6)
Professional and engineering	266 (19.0)	217 (24.9)	49 (9.3)
Clerical	379 (27.1)	261(29.9)	118 (22.4)
Sales	112 (8.0)	46 (5.3)	66 (12.5)
Service	210 (15.0)	91 (10.4)	119 (22.6)
Security	16 (1.1)	14 (1.6)	2 (0.4)
Agriculture, forestry, and fishery	12 (0.9)	5 (0.6)	7 (1.3)
Manufacturing process	107 (7.7)	63 (7.2)	44 (8.4)
Transport and machine operation	25 (1.8)	17 (1.9)	8 (1.5)
Construction and mining	18 (1.3)	16 (1.8)	2 (0.4)
Carrying, cleaning, and packaging	47 (3.4)	9 (1.0)	38 (7.2)
Workers not classified by occupation	105 (7.5)	35 (4.0)	70 (13.3)

**Table 2 ijerph-20-05310-t002:** Descriptive statistics and effect sizes.

	All Workers (*N* = 1398)		Full-Time Workers (*N* = 872)		Part-Time Workers (*N* = 526)		Hedges’ *g* [95% CI]
	*M*	*SD*	*M*	*SD*	*M*	*SD*	
Age (years)	46.09	12.72	44.65	12.37	48.47	12.93	−0.30 [−0.41, −0.19]
Workdays per week	4.92	0.70	5.15	0.48	4.56	0.83	0.93 [0.82, 1.04]
Working hours per week (h)	29.68	13.66	34.44	16.47	21.78	13.66	0.82 [0.71, 0.93]
Commuting time per week (h)	4.87	4.14	5.62	4.35	3.62	3.41	0.50 [0.39, 0.61]
SDI (h)	1.07	1.08	1.11	1.05	1.01	1.11	0.09 [−0.02, 0.20]
WHO-HPQ score	59.94	18.82	59.07	18.35	61.37	19.50	−0.12 [−0.23, −0.01]
PWBS score	89.52	16.44	90.32	15.99	88.18	17.10	0.13 [0.02, 0.24]

Note. *M*, mean; *SD*, standard deviation; CI, confidence interval; SDI, Sleep Debt Index; WHO-HPQ, World Health Organization Health and Work Performance Questionnaire; PWBS, Psychological Well-Being Scale.

**Table 3 ijerph-20-05310-t003:** Correlation matrix for SEM variables (above: full-time workers; below: part-time workers).

	Worktime	Workday	Commuting Time	Sleep Debt	Presenteeism	Well-Being
Worktime	1	0.20 [0.14, 0.26]	0.05 [−0.01, 0.12]	0.01 [−0.06, 0.07]	−0.05 [−0.11, 0.02]	−0.02 [−0.09, 0.05]
Workday	0.31 [0.23, 0.38]	1	−0.02 [−0.08, 0.05]	0.04 [−0.03, 0.10]	−0.01 [−0.07, 0.06]	0.00 [−0.07, 0.07]
Commuting time	0.18 [0.10, 0.26]	0.24 [0.16, 0.32]	1	0.09 [0.02, 0.15]	−0.05 [−0.12, 0.01]	−0.04 [−0.11, 0.03]
Sleep debt	0.02 [−0.07, 0.11]	0.11 [0.02, 0.19]	0.07 [−0.02, 0.15]	1	−0.17 [−0.23, −0.11]	−0.14 [−0.20, −0.07]
Presenteeism	−0.10 [−0.18, −0.01]	−0.09 [−0.17, 0.00]	−0.08 [−0.16, 0.01]	−0.16 [−0.24, −0.08]	1	0.41 [0.35, 0.46]
Well-being	−0.16 [−0.24, −0.08]	−0.05 [−0.13, 0.04]	−0.03 [−0.12, 0.06]	−0.15 [−0.24, −0.07]	0.46 [0.39, 0.53]	1

Note. The values in parentheses indicate a 95% confidence interval.

**Table 4 ijerph-20-05310-t004:** Results of structural equation modeling for all, full-time, and part-time workers.

	Worktime	→	Sleep Debt	Sleep Debt → Presenteeism			Sleep Debt → Well-Being		
	Workday	→	Sleep Debt						
	Commuting Time	→	Sleep Debt						
Group	Estimate	*SE*	β	Estimate	*SE*	β	Estimate	*SE*	β
All workers	−0.001	0.002	−0.013	−2.954	0.461	−0.169 ***	−2.133	0.405	−0.140 ***
	0.111	0.044	0.072 **						
	0.020	0.007	0.077 **						
Full-time workers	−0.000	0.002	−0.006	−2.980	0.581	−0.171 ***	−2.051	0.509	−0.135 ***
	0.082	0.075	0.038						
	0.022	0.008	0.090 **						
Part-time workers	−0.002	0.004	−0.020	−2.817	0.758	−0.160 ***	−2.369	0.665	−0.153 ***
	0.136	0.062	0.102 *						
	0.015	0.015	0.045						

Note. *SE*, standard error; * *p* < 0.05, ** *p* < 0.01, *** *p* < 0.001.

## Data Availability

The data that support the findings of this study are available from the corresponding author upon reasonable request.

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
