# Peer review of "Sleep Debt Mediates the Relationship between Work-Related Social Factors, Presenteeism, and Well-Being in Japanese Workers"

_ijerph, 2023, doi:10.3390/ijerph20075310_

Round 1

Reviewer 1 Report

This paper proposes a study regarding the influence of sleep debt on the relationship between work-related social factors, presenteeism, and well-being. The study was conducted between Japanese  full-time and part-time workers.

The paper is well structured and writtern in good and easy to read English.

The research uses structural equation model (SEM) to test the hypothesys and descriptive statistics for presenting the data. R software was used. The paper also discusses its results and limitation. A wide literature had been uses to compare and discuss the results obtained. The results are concrete and are well presented, being supported by the tables.

The manuscript summarizes the conclusions of the study,  addressing new lines of research and having a  comparative discussion of the authors' findings with those obtained in previous studies in this same area.

Despite the above, the manuscript, in the reviewer's opinion, still has room for improvement regarding the definition of the theoretical framework in the introductory section. That concerns the sleep debt issue, on the one hand, the relationship between work-related social factors, presenteeism, and well-being, on the other, and finally the point of intersection of the topics.

Author Response

Reviewer 1

Thank you for your valuable comments. Our point-by-point responses to your comments are presented below. Revised parts of the manuscript are indicated in blue.

Comment

This paper proposes a study regarding the influence of sleep debt on the relationship between work-related social factors, presenteeism, and well-being. The study was conducted between Japanese full-time and part-time workers. The paper is well structured and writtern in good and easy to read English.

The research uses structural equation model (SEM) to test the hypothesys and descriptive statistics for presenting the data. R software was used. The paper also discusses its results and limitation. A wide literature had been uses to compare and discuss the results obtained. The results are concrete and are well presented, being supported by the tables. The manuscript summarizes the conclusions of the study, addressing new lines of research and having a comparative discussion of the authors' findings with those obtained in previous studies in this same area. Despite the above, the manuscript, in the reviewer's opinion, still has room for improvement regarding the definition of the theoretical framework in the introductory section. That concerns the sleep debt issue, on the one hand, the relationship between work-related social factors, presenteeism, and well-being, on the other, and finally the point of intersection of the topics.

Response

Thank you for your comment. In our revised manuscript, we have added that there is a need for a comprehensive analysis of the association between social factors, sleep debt, presenteeism, and well-being, citing studies published during the COVID-19 pandemic. In addition, we stated in the Conclusion section that our study is important for optimizing worker interventions.

Revised manuscript: page 2, lines 66–75.

The COVID-19 pandemic caused lockdowns worldwide. Initially, it was believed that the lockdown would cause or exacerbate sleep problems, but several studies have re-ported that they would rather increase sleep duration [12, 13]. During the lockdown, most people worked from home, and that condition was associated with worse presenteeism [14]. That is, the results indicate that even with signs of improvement in sleep debt due to lifestyle changes, improvement in job performance may not be im-mediately apparent. Furthermore, no model comprehensively examines work-related social factors, sleep debt, presenteeism, and well-being caused by the lifestyle changes after the outbreak. It is important to understand the relationship between work-related social factors, sleep debt, presenteeism, and well-being to optimize worker interventions.

Revised manuscript: page 10, lines 344–351.

In conclusion, this study captured the increasingly significant social phenomena of commuting time, workday number, sleep debt, presenteeism, and well-being. For the first time, our study found that factors associated with sleep debt differ based on full-time or part-time employment status. The findings showed that reducing working hours does not generally eliminate sleep debt. It means that different employment status requires different methods to address sleep debt. In situations such as the COVID-19 pandemic, where lifestyles change remarkably, the impact of the change on workers’ sleep may likely vary by job type and employment status.

Reviewer 2 Report

Dear Authors,

I enjoyed reading your work. I just have two main issues: 

This idea fits the Japanese context more than other places worldwide. I think it would be great ( if possible ) to suggest if this could be implemented outside Japanese culture or not. 

 Another point: What is the theoretical framework for this research idea?  I suggest that you need to improve the theoretical part. 

Final point: Can you improve the conclusion part? 

Thank you and good luck 

Author Response

Reviewer 2

Thank you for your valuable comments. Our point-by-point responses to your comments are presented below. Revised parts of the manuscript are indicated in blue.

Comment 1

This idea fits the Japanese context more than other places worldwide. I think it would be great ( if possible ) to suggest if this could be implemented outside Japanese culture or not.

Comment 3

Can you improve the conclusion part?

Response

Thank you for your comments. This response captures our responses to Comments 1 and 3. As you pointed out, this study is based on Japanese workers. It seems important to indicate that commuting time and number of workdays were associated with sleep debt, even after controlling for working hours per week. These findings seem beneficial for designing an optimal intervention for sleep debt in workers. We have added this in the Discussion and Conclusion sections.

Revised manuscript: page 9, lines 334–342.

Fourth, this study was conducted on Japanese workers. The relationship between commuting time and sleep debt has been reported in the US [7, 10, 11] and Sweden [31]. On the other hand, the U.S. Bureau of Labor Statistics reported that among Americans, “Other family and/or personal obligations” are the most common reasons for part-time work choices among 25 to 64 years old [32]. In the future, it is necessary to examine whether our model can be replicated in other countries. Fifth, other work-related social factors such as job position, income, and the mode of commuting were not included in this study. In particular, as a commuting mode, the combination of bicycle and walk is associated with higher level of well-being compared to car [33].

Revised manuscript: page 10, lines 344–351.

In conclusion, this study captured the increasingly significant social phenomena of commuting time, workday number, sleep debt, presenteeism, and well-being. For the first time, our study found that factors associated with sleep debt differ based on full-time or part-time employment status. The findings showed that reducing working hours does not generally eliminate sleep debt. It means that different employment status requires different methods to address sleep debt. In situations such as the COVID-19 pandemic, where lifestyles change remarkably, the impact of the change on workers’ sleep may likely vary by job type and employment status.

Comment 2

What is the theoretical framework for this research idea?  I suggest that you need to improve the theoretical part.

Response

Thank you for your comment. In the revised manuscript, we have added that there is a need for a comprehensive analysis of the association between social factors, sleep debt, presenteeism, and well-being, citing studies published during the COVID-19 pandemic.

Revised manuscript: page 2, lines 66–75.

The COVID-19 pandemic caused lockdowns worldwide. Initially, it was believed that the lockdown would cause or exacerbate sleep problems, but several studies have re-ported that they would rather increase sleep duration [12, 13]. During the lockdown, most people worked from home, and that condition was associated with worse presenteeism [14]. That is, the results indicate that even with signs of improvement in sleep debt due to lifestyle changes, improvement in job performance may not be im-mediately apparent. Furthermore, no model comprehensively examines work-related social factors, sleep debt, presenteeism, and well-being caused by the lifestyle changes after the outbreak. It is important to understand the relationship between work-related social factors, sleep debt, presenteeism, and well-being to optimize worker interventions.

Reviewer 3 Report

1. 

"Previous studies have examined the relationships among individual variables but have not comprehensively examined the social phenomenon of work-related social 

factors, presenteeism

---The authors should elaborate on why it is important to consider the social phenemonon. 

2. The authors might as well conduct a power analysis to examine whether the sample size was big enough to detect the effects.

3.The path coefficients showed that an increase in commuting time was significantly associated with an increase in SDI value (? = 0.090, p = 0.008).

---As far as I am concerned, this is just common sense.

4. 

For the first time, our study found that factors associated with sleep 331 debt differ based on full-time or part-time employment status.

---The authors need to elaborate on the implications for this finding.

Author Response

Reviewer 3

Thank you for your valuable comments. Our point-by-point responses to your comments are presented below. Revised parts of the manuscript are indicated in blue.

Comment 1

"Previous studies have examined the relationships among individual variables but have not comprehensively examined the social phenomenon of work-related social factors, presenteeism

---The authors should elaborate on why it is important to consider the social phenemonon.

Response

Thank you for your comment. In the revised manuscript, we have added that there is a need for a comprehensive analysis of the association between social factors, sleep debt, presenteeism, and well-being, citing studies published during the COVID-19 pandemic.

Revised manuscript: page 2, lines 66–75.

The COVID-19 pandemic caused lockdowns worldwide. Initially, it was believed that the lockdown would cause or exacerbate sleep problems, but several studies have re-ported that they would rather increase sleep duration [12, 13]. During the lockdown, most people worked from home, and that condition was associated with worse presenteeism [14]. That is, the results indicate that even with signs of improvement in sleep debt due to lifestyle changes, improvement in job performance may not be im-mediately apparent. Furthermore, no model comprehensively examines work-related social factors, sleep debt, presenteeism, and well-being caused by the lifestyle changes after the outbreak. It is important to understand the relationship between work-related social factors, sleep debt, presenteeism, and well-being to optimize worker interventions.

Comment 2

The authors might as well conduct a power analysis to examine whether the sample size was big enough to detect the effects.

Response

Thank you for your comment. Details of sample size calculation have been added to the manuscript to establish the sufficiency of the study’s sample size. We have added a Statistical analysis section in the revised manuscript.

Revised manuscript: page 5, lines 201–205.

The R package “semTools” (version: 0.5.6) was used to calculate sample size with 6 degrees of freedom, α=0.05, 1-β=0.8. The null hypothesis of RMSEA was set to zero, and the alternative hypothesis was set to 0.08, resulting in a sample size of 356 participants. Therefore, we recruited 356 participants or more in each group.

Comment 3

The path coefficients showed that an increase in commuting time was significantly associated with an increase in SDI value (? = 0.090, p = 0.008).

---As far as I am concerned, this is just common sense.

Comment 4

For the first time, our study found that factors associated with sleep debt differ based on full-time or part-time employment status.

---The authors need to elaborate on the implications for this finding.

Response

Thank you for your useful comments. This response captures our responses to Comments 3 and 4. As you pointed out, it has been shown that commuting time is associated with sleep debt. However, we consider it meaningful to demonstrate that the association between commuting time and number of workdays and sleep debt remains even when work hours per week is controlled for. We have added our thoughts to the Conclusions section of the revised manuscript.

Revised manuscript: page 10, lines 344–351.

In conclusion, this study captured the increasingly significant social phenomena of commuting time, workday number, sleep debt, presenteeism, and well-being. For the first time, our study found that factors associated with sleep debt differ based on full-time or part-time employment status. The findings showed that reducing working hours does not generally eliminate sleep debt. It means that different employment status requires different methods to address sleep debt. In situations such as the COVID-19 pandemic, where lifestyles change remarkably, the impact of the change on workers’ sleep may likely vary by job type and employment status.